# OpenReview forum: "Insights on Disagreement Patterns in Multimodal Safety Perception across Diverse Rater Groups"
_NeurIPS.cc/2024/Workshop/SafeGenAi — SafeGenAi Poster_

### Official Review · Reviewer_Yh2n · 2024-10-09
**Very thorough analysis to statistically uncover latest biases in the human-rated datasets**

**Rating:** 7
**Confidence:** 5

**Review:**

Quality:
The authors highlight a very real problem of aggregated and biased annotated data that can harm both model development and evaluation frameworks.
Detailed mention of how the study was set up, the kinds of images shown to people, and a peek of the UI as shown to the raters
Very thorough detailed statistical analysis with supporting metrics and visual graphs

Clarity:
The challenge and the goal of the project were clearly highlighted in the abstract and then expanded
Key findings were clearly stated and the methodology was clearly defined
The only question about the methodology is why the authors chose to aggregate the annotations per pair into a single rating (lines 154-155) when they began by stating that aggregated ratings mostly obscure the inherent diversity of perspectives in the real-world phenomenon (line 1)


Significance:
Very crucial, at least in today's time when evaluation is more important than anything else
Identifying and uncovering these latent biases that get injected into LLM behaviors needs to be re-calibrated at every level.

---

### Official Review · Reviewer_yz47 · 2024-10-09
**Require Details about the Dataset, Rater Protrait, and Refined Questions**

**Rating:** 4
**Confidence:** 3

**Review:**

This paper investigates the safety issues of generative AI, specifically Text-to-Image models, across different socio-cultural groups. The study focuses on three key features: gender, ethnicity, and age. By examining these aspects, the research provides a unique perspective on safety considerations for future generative AI models across diverse cultural contexts.

Strength:
- The study utilizes a large-scale statistical experiment involving 630 raters and 1000 generated images, providing a robust foundation for analysis.
- The paper offers an in-depth discussion of the survey results, allowing for nuanced insights into the perceptions of safety across different demographic groups.

Weakness:
- Limited Background Information
- Inadequate Explanation of Metrics
- Simplistic Experimental Design

This paper requires a lot of specific domain knowledge, such as the tested dataset and statistical metrics.

The paper would benefit from providing more information about the image dataset used in the study:
- What topics or themes are represented in the generated images?
- What is the overall quality of the images, and how was this determined?
- How diverse is the dataset in terms of content and style?
This context is crucial for readers to fully appreciate the scope and relevance of the study's findings.

The current survey design is somewhat basic. To improve the depth of insights:
- Consider implementing more fine-grained questions that can capture subtle differences in perception across demographic groups.
- Explain the rationale behind the chosen survey questions and how they relate to specific safety concerns in generative AI.

This paper presents an important investigation into the safety perceptions of generative AI across different socio-cultural groups, but it could be significantly strengthened with the aforementioned points.